# Effects of Equine Coaching on Psychoemotional Wellbeing: A Pilot Study in Women with and Without Fibromyalgia

**DOI:** 10.3390/healthcare13212696

**Published:** 2025-10-25

**Authors:** Noelia Rodríguez-Sobrino, Anabel Melguizo-Garín

**Affiliations:** 1Facultad de Ciencias de la Salud, Universidad del Atlántico Medio (UNAM), Tafira Baja, 35017 Las Palmas de Gran Canaria, Spain; 2Facultad de Psicología y Logopedia, Universidad de Málaga (UMA), 29010 Málaga, Spain; anamel@uma.es

**Keywords:** equine coaching, psychoemotional well-being, depression, anxiety, fibromyalgia

## Abstract

Background: Equine-assisted interventions have shown positive effects on psychoemotional well-being. However, little is known about their effects in populations with chronic pain such as fibromyalgia. Objective: This pilot study evaluated the impact of an equine coaching program with and without a diagnosis of fibromyalgia. Methods: The sample consisted of 20 adult women (mean age = 32 years), 12 with fibromyalgia and 8 without a clinical diagnosis. Instruments used included the Beck Depression Inventory-II (BDI-II), the State-Trait Anxiety Inventory (STAI), and the General Health Questionnaire-28 (GHQ-28). Results: The findings indicated significant improvements in depression, anxiety, and general health in both groups, suggesting benefits for women with and without fibromyalgia. Conclusions: These findings suggest potential benefits of equine coaching as a complementary approach to psychoemotional wellbeing, although causal conclusions cannot be drawn.

## 1. Introduction

Animal-assisted therapies, and in particular those incorporating horses, have attracted increasing interest in recent years for their potential in the field of psychoemotional well-being. Numerous studies suggest that horses act as “emotional mirrors”, offering immediate feedback that facilitates the development of emotional awareness and self-regulation, key elements in therapeutic contexts [1,2,3,4].

Equine coaching or equine-assisted therapy (EAT) is based on experiential interaction with horses to promote positive change in psychoemotional well-being [5,6].

Although the empirical evidence is still emerging, several papers have documented the benefits of EAT in reducing symptoms of anxiety, depression, and post-traumatic stress [7,8,9,10,11,12,13,14,15,16,17], as well as in strengthening resilience, emotional regulation and stress coping [18,19]. A positive impact on physiological markers such as decreased cortisol has also been reported [20], although findings in this area are still mixed [21].

With regard to depression, previous studies have shown significant symptom improvement in people who participated in equine coaching programmes [22,23,24,25,26]. This modality has also shown promising results in vulnerable populations, such as adolescents with suicidal ideation or students with a high academic load [18,27].

The psychosocial impact of EAT seems particularly relevant in contexts where emotional disorders coexist with chronic medical conditions, as in the case of fibromyalgia (FM). This disease, characterised by chronic widespread pain, fatigue and involvement of the nervous, endocrine and immune systems, has a high prevalence of comorbidity with anxiety and depression [28,29,30,31]. Women with FM, in particular, report a marked deterioration in their quality of life, making it necessary to explore complementary interventions that comprehensively address their emotional well-being.

The present study aims to analyse the effect of an equine coaching programme on well-being, specifically assessing changes in depression, anxiety and perception of general health.

Through a quasi-experimental design with groups differentiated by fibromyalgia diagnosis, the aim was to explore the efficacy of the intervention in both clinical and non-clinical populations. We also examined the possible lower impact on trait anxiety, considered a more stable dimension of personality.

## 2. Method

### 2.1. Participants

The sample consisted of 20 adult women, residents of the island of Gran Canaria, Spain.

### 2.2. Instruments

A quantitative methodology was used, with evaluation at two points in time: before the start of the intervention (pretest) and after the eighth session of equine coaching (posttest). The battery of instruments was applied in paper format, uniformly throughout the sample, and corrected according to the authors’ guidelines. The following standardized and validated questionnaires were used:

**The Beck Depression Inventory–II (BDI-II)** is a 21-item self-report measure that is utilized to assess the severity of depressive symptoms. Each item is evaluated using a 4-point Likert scale (0–3), resulting in a total score ranging from 0 to 63. According to the Spanish adaptation [32], the cut-off scores are as follows: minimal (0–13), mild (14–19), moderate (20–28), and severe (29–63) depression.

**The State-Trait Anxiety Inventory (STAI) [33]** is a widely validated instrument for the assessment of anxiety in both clinical and general populations, including individuals suffering from chronic pain and fibromyalgia. The scale is composed of two subscales of 20 items each. These items are rated on a 4-point Likert scale (1–4). Scores for each subscale range from 20 to 80, with higher scores indicating greater anxiety. Interpretation followed the cut-off values established for the Spanish population [34].

**The General Health Questionnaire-28 (GHQ-28)** is an instrument which groups 28 items into four subscales (somatic symptoms, anxiety/insomnia, social dysfunction, and depressive symptoms). Each item is scored using the GHQ method (0-0-1-1). The cut-off point of 5/6, proposed by [35] for identifying potential psychiatric cases, was applied. Scores range from 0 to 28, with higher values indicating poorer perceived general health.

### 2.3. Procedure

The study, which was of a pilot and exploratory nature, was approved by the Ethical Committee and the intervention was developed in collaboration with the organization Tierra de Caballos (Arucas, Spain). The intervention was coordinated by the principal investigator (psychologist), with the support of two facilitators and a psychology trainee. Twelve horses living in herd and semi-freedom conditions were used.

Participants were recruited from those who had previously applied for equine coaching at Tierra de Caballos and met the inclusion criteria. After an initial briefing, they were given informed consent and data protection documentation to sign. Assignment to the experimental group (diagnosis of fibromyalgia) or comparison group (no clinical diagnosis) was made, and the intervention sessions were planned. The protocol was developed in three phases. In phase 1 (pre-test), assessment instruments were administered to measure depression, anxiety and general health. In phase 2, both conditions participated in eight weekly intervention sessions (2 h each), outdoors and in direct contact with the herd. The sessions took place between June and December 2023, adapting to the participants’ personal schedule. In phase 3 (post-test), the battery of instruments was re-administered to evaluate the impact of the intervention.

The intervention consisted of eight weekly sessions (2 h each) focused on experiential interaction with the horses, emotional regulation, body awareness, and communication. Each session followed a standardized structure including a brief mindfulness exercise, guided horse interaction, and group reflection. All facilitators adhered to a shared intervention protocol supervised by the principal investigator to ensure methodological consistency. The resources and methodologies applied in the sessions were as follows:-Facilitators: 2 coaches + 1 psychologist-Horses: 12 horses that live in a herd and in semi-free conditions-Assessment tests (psychometric tests)-Methodologies applied:-Horse-assisted coaching-Systemic movement (family constellations)-Cardiac coherence (with biofeedback measurement using Inner Balance devices)-Mindfulness

The standardized protocol for each session was as follows:Begin by defining and/or specifying the working objective of the session.Conduct awareness-raising activities and exercises are carried out: by observing the behavior of the horses (which act as a mirror of each participant’s emotions) during the exercise, the facilitator asks questions that invite reflection.Provide support and tools to navigate and manage emotions during the process.Conclusions are drawn from the session and objectives are defined. The session ends with the establishment of new commitments.

### 2.4. Statistical Analysis

For data analysis, inferential statistics with *t*-tests for related samples were used to compare levels of depression, anxiety (state and trait) and general health before and after the intervention in each group. Differences were tested for statistical significance (*p* < 0.05) and effect sizes were calculated using Cohen’s *d*, in order to estimate the magnitude of change.

In addition to paired-samples *t*-tests, a Multivariate Analysis of Variance (MANOVA) was performed to evaluate the overall multivariate effect of the intervention (pre–post) and the group factor (experimental vs. control). This approach allowed assessing the simultaneous effect of the intervention on depression, anxiety (state and trait), and general health. Prior to analysis, assumptions of normality, homogeneity, and multicollinearity were tested.

No formal correction for multiple comparisons (e.g., Bonferroni adjustment) was applied due to the exploratory nature of this pilot study. Consequently, reported *p*-values and effect sizes should be interpreted with caution.

All statistical analyses were performed using IBM SPSS Statistics for Windows, version 25.0 (IBM Corp., Armonk, NY, USA).

## 3. Results

This section presents the main results obtained from the data analysis, highlighting the most relevant trends and the relationships identified between the studied variables. All of them voluntarily agreed to participate in the study and signed the informed consent form. The inclusion criteria were:Being 18 years of age or older.Present a medical diagnosis of fibromyalgia (experimental group).Accepting and signing the informed consent form.

Exclusion criteria included:Neuropsychiatric diagnosis.Medical diagnosis other than fibromyalgia or current or recent psychological treatment.Age under 18 years.Refusal or non-signing of informed consent.

### 3.1. Participant Characteristics

The final sample consisted of 20 adult women, with an overall mean age of (M = 32.4 years (SD = 7.1; range = 22–47). Twelve participants were included in the experimental group (fibromyalgia diagnosis) and eight in the comparison group (no previous diagnosis). All participants were residents of Gran Canaria and shared similar sociodemographic characteristics (middle socioeconomic level, higher education in 65% of cases). No participant dropped out during the intervention.

### 3.2. Descriptive Statistics and Psychometric Results

Table 1 displays the descriptive statistics and effect sizes (Cohen’s d) for all study variables in the experimental and comparison groups at pre-test and post-test. The measures included Depression (BDI-II), State and Trait Anxiety (STAI-E and STAI-R), and General Health (GHQ-28) with its four subscales.

The results obtained for the variables assessed (depression, anxiety and general health) are presented, differentiating between the experimental and comparison groups. For both groups, pre-post comparisons were performed using *t*-tests for related samples. In addition, a multivariate analysis was performed to examine the overall effect of the intervention and the group of belonging.

Results indicated a significant improvement across time in both the experimental and comparison groups, reflected by reductions in depressive and anxiety symptoms and better general health scores, with large effect sizes observed in most measures.

In the experimental group, composed of participants diagnosed with fibromyalgia, the analysis revealed a significant reduction in three of the four variables assessed. Specifically, Depression (BDI-II) and State Anxiety (STAI-E) showed marked decreases, while Trait Anxiety (STAI-R) also declined, approaching statistical significance. Regarding General Health (GHQ-28), the improvement was particularly notable, with participants moving from clinical to non-clinical levels of psychological distress after the intervention.

In the comparison group, composed of participants without a clinical diagnosis, statistically significant reductions were observed across all the assessed variables Participants showed lower levels of Depression (BDI-II), State Anxiety (STAI-E), and Trait Anxiety (STAI-R), together with a substantial improvement in General Health (GHQ-28), reflecting an overall enhancement in psychological well-being following the intervention.

Multivariate analysis (MANOVA) showed a significant effect of the time factor (pre-post), indicating that the intervention had an overall impact on the variables evaluated (*Wilks’ Lambda* = 0.045, *F*(8,11) = 29.27, *p* < 0.001). However, no significant differences were found as a function of group (experimental vs. comparison), (*Wilks’ Lambda* = 0.847, *F*(8,11) = 0.25, *p* = 0.971) (see Table 2). This suggests that equine coaching was beneficial in both groups, regardless of previous diagnosis.

The findings show a significant decrease in levels of depression, state anxiety and general health symptoms in both the experimental and comparison groups following the equine coaching intervention. These reductions suggest an improvement in the psychoemotional well-being of the participants, regardless of their previous diagnosis. No significant differences were observed between groups, supporting the usefulness of this intervention in both clinical and non-clinical settings.

## 4. Discussion and Conclusions

The results of this pilot study allow us to respond to the main objective: to analyse the effect of an equine coaching programme (EAT) on psychoemotional well-being in women with and without a diagnosis of fibromyalgia. The data show a significant reduction in depression, state anxiety and general health scores after the intervention, both in the experimental group and in the comparison group. The improvement is observed in both groups, indicating that EAT can be effective regardless of the presence of a prior clinical diagnosis.

A recent systematic review [36] has also highlighted the emerging evidence supporting equine-assisted programs in improving emotional regulation and stress reduction, although methodological variability remains high across studies.

The baseline analysis revealed that participants in the fibromyalgia group presented higher levels of depression, anxiety, and poorer general health than those in the comparison group. These findings are consistent with previous research on fibromyalgia populations [28,29,31], where elevated emotional distress is commonly observed. Although both groups improved significantly after the intervention, these initial imbalances may have partially influenced the magnitude of change. Future studies with larger samples should consider controlling for baseline differences using a multivariate covariance model (MANCOVA).

In line with previous research [8,18,24], this study provides empirical evidence of the potential of equine-assisted interventions to improve emotional well-being and reduce psychological distress. In particular, the results show a significant decrease in depression and state anxiety, while trait anxiety, although showing a less pronounced decrease, also reflected some sensitivity to treatment, suggesting a possible regulatory effect even on more stable dimensions of emotional functioning. Multivariate analysis confirms the significant effect of time (pre-post) on all variables assessed, with no differences between groups, reinforcing the usefulness of equine coaching as a complementary intervention in both clinical and non-clinical populations. Furthermore, the observed effect sizes were high in both conditions, indicating a clinically relevant magnitude of the changes obtained.

However, this study has some limitations that should be taken into account. First, the sample size is small and homogeneous (adult women living in the same region), which limits the generalisability of the findings. Secondly, a non-intervention comparison group was not included, which would have allowed us to control more precisely for the specific effect of EAT versus other external factors. There was also no longitudinal follow-up, so the stability of the effects over time cannot be established. Finally, although validated instruments (BDI-II, STAI, GHQ-28) were used, no other physiological or observational measures were incorporated to complement the self-reported perspective.

The marked improvement observed in both groups, including the comparison group, may partially reflect expectancy effects, regression to the mean, or contextual influences associated with participation in an emotionally supportive, outdoor activity [37]. These factors should be considered when interpreting the results, as they may have contributed to the observed psychological gains.

The limitations now explicitly include small sample size, absence of a non-intervention group, potential self-selection bias, and uncorrected multiple testing. Future research with larger and more diverse samples could apply more robust statistical models to verify these preliminary findings.

Despite these limitations, the results reinforce the value of EAT as a low-risk, accessible intervention with therapeutic potential, especially useful in contexts where traditional verbal intervention may be limited. These findings open up promising avenues for future research. It would be desirable to replicate this study with larger samples, to include diverse populations (by age, gender, clinical situation), and to explore different intervention formats (duration, frequency, number of sessions). Longitudinal studies would make it possible to examine the sustainability of the effects in the medium and long term. The incorporation of biological or relational indicators could also contribute to a better understanding of the mechanisms of change involved in EAT.

In addition, two further limitations should be noted. First, a potential participation effect may have contributed to the observed improvements, as involvement in a structured research activity could have had positive psychosocial effects. Second, participants’ prior interest in equine-assisted activities, as they had voluntarily applied for equine coaching before recruitment, could have introduced an expectancy bias that partially explains the positive outcomes.

In conclusion, preliminary evidence is provided on the efficacy of the EAT as a tool for improving variables such as depression, anxiety and general health. The absence of intergroup differences suggests that it may be useful in both clinical and undiagnosed populations. It is suggested to continue with this line of research on the development of the EAT as a standardized, innovative and complementary practice for health and psychoemotional well-being programmes.

## Figures and Tables

**Table 1 healthcare-13-02696-t001:** Psychological outcomes before and after intervention.

Variable	Group	Pre-Test (M ± SD)	Post-Test (M ± SD)	Range	Cohen’s *d*
BDI-II (Depression)	Experimental	25.92 (11.19)	14.33 (10.28)	0–63	1.08
	Comparison	49.00 (15.64)	27.00 (10.30)	0–63	1.66
STAI-E (State Anxiety)	Experimental	34.50 (8.76)	17.75 (12.15)	20–80	1.58
	Comparison	47.00 (11.84)	33.00 (9.62)	20–80	1.30
STAI-R (Trait Anxiety)	Experimental	38.17 (9.67)	28.17 (11.07)	20–80	0.96
	Comparison	56.00 (14.94)	42.00 (11.60)	20–80	1.05
General Health (GHQ-28)	Experimental	16.75 (6.24)	5.58 (5.95)	0–28	1.83
	Comparison	25.00 (9.32)	11.00 (4.09)	0–28	1.95
GHQ Subscales					
Somatic Symptoms	Experimental	5.3 (2.1)	2.1 (1.8)	0–7	–
Anxiety/Insomnia	Experimental	4.8 (2.0)	1.9 (1.5)	0–7	–
Social Dysfunction	Experimental	3.9 (1.7)	1.3 (1.2)	0–7	–
Depressive Symptoms	Experimental	2.8 (1.9)	0.9 (1.0)	0–7	–

**Note**. The inclusion of subscale means provides a more detailed view of health-related changes.

**Table 2 healthcare-13-02696-t002:** Results of multivariate analysis (MANOVA) for the effect of time (pre-post) and group (experimental vs. comparison).

Effect	Statistic	Value	F	gl (Hypothesis)	gl (error)	Sig (*p*)
Intersection	Pillai trace	0.96	29.27	8	11	0.00
	Wilks Lambda	0.05	29.27	8	11	0.00
	Hotelling trace	21.28	29.27	8	11	0.00
	Roy’s major root	21.28	29.27	8	11	0.00
Group	Pillai Trace	0.15	0.25	8	11	0.97
	Wilks’ Lambda	0.85	0.25	8	11	0.97
	Hotelling trace	0.18	0.25	8	11	0.97
	Roy’s root greater than	181	0.25	8	11	0.97

**Note**. A design with intersection (pre-post time) and group (experimental vs. comparison) factors was used. All statistics are exact. Before applying inferential analyses, assumptions of normality of differences were checked using the Shapiro–Wilk test, with no significant deviations detected. This allowed the use of *t*-tests for related samples. In addition to statistical significance values, 95% confidence intervals were calculated for pre-post mean differences, which do not include zero for those variables with significant changes, reinforcing the reliability of the results.

## Data Availability

The data are not publicly available due to privacy and ethical restrictions. Data are available on request from the corresponding author.

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
