# Peer review of "Effects of Equine Coaching on Psychoemotional Wellbeing: A Pilot Study in Women with and Without Fibromyalgia"

_healthcare, 2025, doi:10.3390/healthcare13212696_

Round 1

Reviewer 1 Report

Comments and Suggestions for Authors

I would like to thank the authors for the opportunity to read this very interesting and nicely written manuscript. I appreciate the overall premise of the study and I am glad to see such research being conducted.

The manuscript details a study investigating the use of equine-assisted therapy to improve psychoemotional wellbeing in women with fibromyalgia. Women without fibromyalgia were included as controls. It was framed as a pilot study which is appropriate given the small sample size. A number of psychometrics were employed as outcome measures (Beck Depression Inventory-II, State-Trait Anxiety Questionnaire, and General Health Questionnaire-28); and t-tests then multivariate MANOVA were used to assess the differences between the experimental and the control groups. Post-intervention both groups demonstrated significant improvement across all measures compared to pre-intervention levels, however, no significant difference was seen between the groups. The authors concluded that equine-assisted therapy may be useful for improving depression, anxiety and general health psychometric outcomes in broader populations.  

However, there some areas of concern to be addressed before it can be recommended for publication.  

Methods

The description of the participants includes details that would be better placed at the start of the results section: the number of participants included in total/in each group, and the mean age(s).

The description of the measurement instruments lack detail, for example:
Beck Depression Inventory-II: while Likert-items are mentioned the scale is not (4-point).
State-trait Anxiety Questionnaire: there is no detail of the Likert-scale, range of scores, or how the scores are interpreted (thresholds). There is also no associated reference.
General Health Questionnaire: while the scoring system is included (0, 0, 1, 1) there is no detail of the methodology behind it. This is the same for the cut-off point of 5/6. The score range and its interpretation are also missing.

While MANOVA multivariate statistical analysis is reported in the results section, it is absent from the Statistical analysis section.

Results

There are no descriptive details of the included participants. Things that might be helpful to the reader include: age ranges overall and for each group; age mean for each group, any further demographics (if collected).

No detail is provided around the collected psychometric data. Examples of what might be of interest to the reader are:
Overall and group results with regard to their depression, state/trait anxiety and general health (including general health subscales [somatic symptoms, anxiety/insomnia, social dysfunction and depressive symptoms]) score ranges and threshold distributions.
-A table might assist with communicating this detail.

Suggest this is of particular interest given the groups appear to be somewhat different with regard to their depression, anxiety and general health scores at baseline.

Also, given the substantial differences between the groups at baseline for all measures, suggest using the MANCOVA as your multivariate test as it specifically controls for baseline scores to account for any potential imbalances between groups.

Discussion

There is no discussion around the baseline psychometric scores of the groups and what that might mean for their depression, anxiety and general health.
Also, no discussion around how the depression, anxiety and general health scores in the fibromyalgia group compared with previously reported fibromyalgia studies reporting depression, anxiety and general health outcomes.

There is no discussion about the baseline differences in depression, anxiety and general health between the groups and how that might have impacted the outcomes of the study. This might be best placed as a limitation.

Further limitations
No discussion of other factors that may have contributed to the significant results in both groups, for example research participation effects.
No discussion that the recruited participants were those who had previously applied for equine coaching therefore, it is reasonable to think they may have had a pre-exiting enthusiasm for the activity which may have biased the study outcomes.

Comments on the Quality of English Language

Introduction

The last paragraph is written in the present tense however, it details actions that have been completed, for example “Through a quasi-experimental design with groups differentiated by fibromyalgia diagnosis, the aim is [was] to explore the efficacy of the intervention in both clinical and non-clinical populations. We also examine[d] the possible lower impact on trait anxiety, considered a more stable dimension of personality.”

Reviewer 2 Report

Comments and Suggestions for Authors

This manuscript explores a novel and increasingly relevant topic with particular focus on women with and without fibromyalgia. The study is clearly written, employs validated instruments, and provides encouraging preliminary findings. The work has merit and could be of interest to readers of Healthcare. However, several methodological and interpretive issues should be addressed before it can be considered for publication.

My major concerns:

  1. Control Group Design
    Both groups received the intervention, one with and one without fibromyalgia. This is not a true control group, and the terminology may be misleading. The absence of a no-intervention or alternative-treatment arm limits causal inference and should be emphasised more strongly.

  2. Sample Size and Power
    With only 20 participants, divided unevenly, the study is underpowered for meaningful between-group comparisons. The authors should present the findings as exploratory and avoid overinterpretation.

  3. Statistical Considerations
    Multiple t-tests increase risk of Type I error. Were any corrections (e.g., Bonferroni) considered? Reported large effect sizes should be interpreted with caution given the small n and lack of control.

  4. Interpretation of Results
    The marked improvements in the so-called control group are noteworthy but raise questions about expectancy effects, regression to the mean, or contextual influences. This deserves more discussion. The conclusion that equine coaching is equally effective regardless of diagnosis should be toned down.

My minor concerns

  • The abstract should explicitly mention the absence of a non-intervention comparison group.

  • Recruitment strategy (self-selection) may introduce bias; this should be acknowledged.

  • More detail on session content and standardisation across facilitators would strengthen reproducibility.

  • Tables could be reformatted for clarity (e.g., clearer alignment of effect sizes).

  • The discussion would benefit from referencing additional systematic reviews to situate the findings within the broader evidence base.

Reviewer 3 Report

Comments and Suggestions for Authors

Dear Authors,
I had the opportunity to review your article, which tested, in pilot study form, the usefulness of equine-assisted interventions in improving the psychoemotional well-being of both pathological (fibromyalgia) and healthy adult women. Your results seem to indicate that the proposed intervention is useful for improving psychoemotional well-being regardless of the patient's pathological status.

GENERAL ASPECTS
- The article is extremely short to be presented as an "Article." Since it is a rather concise pilot study, I would suggest presenting it as a "Brief Report." This would not affect the paper's value, but would better contextualize it in the selected journal.
- In-text citations are made with the author's name and publication year in parentheses. For clarity and flow, please follow the MDPI format (numbered citations in the main text, with the number in square brackets).

TITLE
- Okay, it correctly defines the study design and target population.

ABSTRACT
- Lines 8-9: There is a lack of connection between the premise ("Equine-assisted interventions have shown positive effects on psychoemotional well-being") and the study design ("This pilot study evaluates the impact of an equine coaching program..."). You should indicate what led you to investigate this approach in comparing healthy and fibromyalgia patients (example: there is no specific literature on equine coaching in fibromyalgia).
- Line 10: "The sample consisted of 20 adult participants." It would be better to write "The sample consisted of 20 adult women (mean age 32 years)."
- Line 12: "BDI-II, STAI, and GHQ-28." The acronyms should be spelled out in full in the abstract to avoid confusing the reader.
- Lines 15-16: "...suggesting that the benefits of equine coaching are not dependent on prior diagnosis." It would be better not to generalize so much and simply say that equine coaching is beneficial for both fibromyalgia patients and healthy women, without referring too much to other diagnoses not directly investigated.

INTRODUCTION
- Line 27: “... to promote positive change,” changes in what? Perhaps you should specify “in psychoemotional well-being.”
- The content, in general, is acceptably clear.

METHODS:
- Lines 59-60: “residents of the island of Gran Canaria” please also indicate the nation (I guess it’s Spain, but you should indicate the reference nation whenever you also indicate a place, to avoid confusion).
- Line 84: “State-Trait Anxiety Questionnaire (STAI)...” a bibliographic reference on the usefulness of this tool for the studied pathological contexts would be helpful (as you correctly did for the other two scales).
- Line 97: “Tierra de Caballos (Arucas)” again, indicate the nation.
- Line 107: “In phase 2...” A major problem is that you don't clearly indicate what the approach consisted of! Considering that this is a pilot study, which already has methodological limitations, at the very least, to enhance its value, it would be helpful to clearly indicate what the EAT consisted of (i.e., what exactly did the subjects do during each equine coaching session?).
- line 112: “Statistical analysis” you should indicate which calculation program was used to conduct the selected tests (Program name, software company name, and country).

RESULTS:
- In general, it makes no sense to repeat the results of the individual groups twice, once in a conversational manner and then in a table (Table 1). Tables are more than clear and sufficient, and the textual presentation of the results at the beginning of the results section is redundant.
- Lines 120-121: “For both groups, pre-post comparisons were performed using t-tests for related samples.” You already said this in the previous paragraph, so there's no need to repeat it.
- Line 160: "Multivariate analysis (MANOVA)...", isntead, why didn't you indicate that the group comparison was performed using this method in section "2.4. Statistical analysis."? Please indicate so there.

DISCUSSION & CONCLUSIONS
- in general ok

ACCESSORY SECTIONS:
- Several mandatory MDPI statements are missing at the bottom of the text (funding, conflict of interest, ethics, etc.). Please complete and insert them.

In conclusion, I consider your article quite interesting and full of potential. However, there are several shortcomings both in the paper's structural organization and in the details of the applied methodology. Therefore, I recommend that you make the suggested changes with a major revision, hoping to publish this project as soon as possible. Good luck with your work.

Round 2

Reviewer 1 Report

Comments and Suggestions for Authors

I appreciate the authors efforts in responding to the review comments and the manuscript is much improved for it. There are still some issues that will need to be resolved before acceptance for publication.

Methods:

Lines 57-60. As per my previous review this information does not belong in the methods section, this is your results.  

Lines 71-76. Please correct this section for English grammar, and readability; and reference formatting.

Line 80: Please correct the reference formatting.

Lines 135-137. The MANCOVA sentence does not belong in the methods section in belongs in the Discussion section. Currently you have referred to it in the general Discussion section, but the text "Given the exploratory and pilot nature of the study, MANCOVA was not applied to avoid model overfitting" belongs in the Discussion limitations section. 

Lines 138-140. This text also belongs in the limitations section of the discussion.

Results:

Lines 147-156. This text belongs in the Methods section under the ‘Participants’ heading.

Lines 171-172. You have hyphenated words when it is not appropriate.  

Line 176. Your results indicated improvement across time in both the experimental group and the control group.

Discussion:

Lines 251-255. Please provide references in support of these claims.

Lines 256-257. This text reads like comments to the reviewer rather than text for the manuscript.

Comments on the Quality of English Language

Some typos, grammar and readability issues in need of correction that have been included in the reviewer comments 

Reviewer 2 Report

Comments and Suggestions for Authors

I believe the manuscirpt can now be published. 

Author Response

Response (English): 
We sincerely thank the reviewer for their positive evaluation and for recommending our manuscript for publication. We appreciate the constructive feedback provided throughout the review process, which greatly contributed to improving the quality and clarity of the paper. 

Reviewer 3 Report

Comments and Suggestions for Authors

The article describing this form of equine-supported treatment is significantly improved compared to the originally submitted version, which I reviewed in the past.
The "abstract" has been reorganized to be clearer and more consistent with journal standards.
The "introduction" contains some additional specifications.
The "methods" section is much clearer, including more extensive descriptions of the assessment tools, the treatment protocol, and the analysis methods used.
The "results" section is much more extensive and detailed, and includes new tables reflecting the new data analyses.
The "discussion" has been slightly expanded and is consistent with the results.
The "mandatory statements" have been added at the end of the text.
I consider the authors' work to be very good and commendable in improving the paper.
I would, however, recommend reviewing the entire text for typos or misprints (e.g., residual deletions or incorrectly spaced words).
I wish you good luck with your work.

Author Response

Response (English): 
We sincerely thank the reviewer for their positive and encouraging feedback. We are pleased that the revised version meets the journal’s standards and that the improvements in clarity, methodological detail, and structure have been appreciated. Following the reviewer’s final suggestion, we have carefully reviewed the entire manuscript for typos, spacing inconsistencies, and minor formatting issues to ensure accuracy and readability in the final version.